A new species of the odorous frog genus Odorrana (Amphibia, Anura, Ranidae) from southwestern China

Li Shize 1 2
Xu Ning 1
Lv Jingcai 3
Jiang Jianping 2
Wei Gang 4 wg198553@126.com
Wang Bin 2 wangbin@cib.ac.cn
1 Department of Food Science and Engineering, Maotai University , Renhuai, Guizhou , China
2 CAS Key Laboratory of Mountain Ecological Restoration and Bioresource Utilization & Ecological Restoration and Biodiversity Conservation Key Laboratory of Sichuan Province, Chengdu Institute of Biology, Chinese Academy of Sciences , Chengdu, Sichuan , China
3 Guizhou Institute of Biology, Guizhou Academy of Sciences , Guiyang, Guizhou , China
4 Biodiversity Conservation Key Laboratory, Guiyang College , Guiyang, Guizhou , China
Wink Michael
Electronic publication date: 2018 Oct 4
Publication date: 2018
Volume: 6
Electronic Location ID: e5695
Received 2018 Apr 20; Accepted 2018 Sep 5
Copyright: © 2018 Li et al.
Copyright year: 2018
Copyright holder: Li et al.
License: This is an open access article distributed under the terms of the Creative Commons Attribution License, which permits unrestricted use, distribution, reproduction and adaptation in any medium and for any purpose provided that it is properly attributed. For attribution, the original author(s), title, publication source (PeerJ) and either DOI or URL of the article must be cited.
License URL: https://creativecommons.org/licenses/by/4.0/

Keywords: Taxonomy, Odorrana kweichowensis sp. nov., Phylogenetic analyses, Mitochondrial DNA, Nuclear DNA, Morphology, Southwestern China

Funding: Strategic Priority Research Program of the Chinese Academy of Sciences XDB31000000 National Key Research and Development Program of China 2017YFC0505202 National Natural Sciences Foundation of China NSFC-31360144 and NSFC-31201702 Biodiversity Conservation Key Laboratory of Guizhou province Education Department, Guiyang College; the laboratory on biodiversity conservation and applied ecology of Guiyang college; Ocean Park Conservation Foundation, Hong Kong PR 1030001252 This study was supported by the following foundations: the Strategic Priority Research Program of the Chinese Academy of Sciences, Grant No. XDB31000000; the National Key Research and Development Program of China (No. 2017YFC0505202); the National Natural Sciences Foundation of China (Nos. NSFC-31360144 and NSFC-31201702); Biodiversity Conservation Key Laboratory of Guizhou province Education Department, Guiyang College; the laboratory on biodiversity conservation and applied ecology of Guiyang college; Ocean Park Conservation Foundation, Hong Kong (No. PR 1030001252). The funders had no role in study design, data collection and analysis, decision to publish, or preparation of the manuscript.

==============================
The genus Odorrana is widely distributed in the mountains of East and Southeastern Asia. An increasing number of new species in the genus have been recognized especially in the last decade. Phylogenetic studies of the O. schmackeri species complex with wide distributional range also revealed several cryptic species. Here, we describe a new species in the species complex from Guizhou Province of China. Phylogenetic analyses based on mitochondrial DNA indicated the new species as a monophyly clustered into the Odorrana clade and sister to O. schmackeri, and nuclear DNA also indicated it as an independent lineage separated from its related species. Morphologically, the new species can be distinguished from its congeners based on a combination of the following characters: (1) having smaller body size in males (snout-vent length (SVL) <43.3 mm); (2) head longer than wide; (3) dorsolateral folds absent; (4) tympanum of males large and distinct, tympanum diameter twice as long as width of distal phalanx of finger III; (5) two metacarpal tubercles; (6) relative finger lengths: II < I < IV < III; (7) tibiotarsal articulation reaching to the level between eye to nostril when leg stretched forward; (8) disks on digits with circum-marginal grooves; (9) toes fully webbed to disks; (10) the first subarticular tubercle on fingers weak; (11) having white pectoral spinules, paired subgular vocal sacs located at corners of throat, light yellow nuptial pad on the first finger in males.

Introduction

Frogs of the genus Odorrana Fei, Ye & Huang, 1990 inhabit montane streams in the subtropical and tropical regions of East and Southeast Asia, ranging from the Ryukyu Archipelago of southern Japan, southern China and Indochina, northeastern India, Myanmar and Malay Peninsula to Sumatra and Borneo (Frost, 2018). Although the systematic relationships of the group had been controversial for decades, it has been recognized as a monophyly in recent years (Matsui et al., 2005; Ngo et al., 2006; Cai et al., 2007; Che et al., 2007; Stuart, 2008; Wiens et al., 2009; Pyron & Wiens, 2011; Chen et al., 2013). The genus currently contains 58 recognized species (Frost, 2018). Noticeably, in the last decade, 13 species have been described (Li, Lu & Rao, 2008; Tran, Orlov & Nguyen, 2008; Yang, 2008; Bain et al., 2009; Chen, Zhou & Zheng, 2010a, 2010b; Kuramoto et al., 2011; Mo et al., 2015; Wang et al., 2015; Pham et al., 2016; Saikia, Sinha & Kharkongor, 2017). This indicated that the species diversity has been underestimated and more discoveries were obligatory in the genus.

The piebald odorous frog O. schmackeri was firstly found in Gaojiayan Town, Changyang County (Co.), Hubei province (Prov.), China, and named by Boettger (1892) only based on one male specimen and with a negligible description. Thus, Liu & Hu (1961) redescribed O. schmackeri based on some specimens from Daba Mountains and Xiushan Co. in Chongqing City of China near to the type locality of the species, and presented the diagnosis characters for the species, such as having great circle brown spots on dorsa and the outer vocal sac below pharynx in males. Afterward, based on mass of reports and investigations, the species has been suggested to be distributed in Henan, Shaanxi, Gansu, Sichuan, Guizhou, Hubei, Anhui, Jiangsu, Zhejiang, Hunan, Guangdong and Guangxi provinces of China (Fei et al., 2009; Fei, Ye & Jiang, 2012), even northern Thailand (Chan-ard, 2003) and northern Vietnam (Orlov et al., 2002; Nguyen, Ho & Nguyen, 2005). This group is highly adapted to mountain environments, inhabiting the moist evergreen broad-leaf forests and streams at 200–1,400 m altitudes (Fei et al., 2009). Accordingly, it could be assumed that the wide distributional range and strict habitat requirements might promote considerable divergence even speciation due to isolation in the group. Correspondingly, populations ever classified as O. schmackeri were suggested to be much variable in morphology (Fei et al., 2009; Fei, Ye & Jiang, 2012), indicating that there might be cryptic species in the group. Indeed, from this group, several new species have been constantly described: O. nanjiangensis Fei, Ye & Jiang, 2007 occurring from Nanjiang County (Co.), Sichuan Province (Prov.); O. huanggangensis Chen, Zhou & Zheng, 2010a occurring from Wuyi Mountains in Fujian and Jiangxi provinces; and O. tianmuii Chen, Zhou & Zheng, 2010b occurring from Tianmu Mountains in Zhejiang Prov. However, phylogenetic analyses of Chen et al. (2013) indicated that above four species did not form a monophyly, but were nested with some other Odorrana species including two cryptic species, and also strongly rejected the alliance of O. anlungensis, O. yizhangensis, and O. lungshengensis with the O. schmacheri group that was defined based on morphological data (Fei et al., 2009). Therefore, the “O. schmackeri species complex” was defined referring to the species resembling O. schmackeri on morphology, such as O. nanjiangensis, O. huanggangensis, O. hejiangensis, O. tianmuii, and several cryptic species (Li et al., 2015; Zhu, 2016; He, 2017).

Recently, many studies have paid attention to the phylogenetic relationships and diversification of the O. schmackeri species complex (Li et al., 2015; Zhu, 2016). Li et al. (2015) presented a phylogeographic framework for the species complex using 25 populations based on mitochondrial ND2 and two tRNA gene sequences, and proposed seven clades: Clade A was (O. huanggangensis + “O. yizhangensis” identified by them) from Wuyi Mountains in Fujian Prov., Nanling Mountains in the border of Guangdong and Hunan provinces and mountains in eastern Guizhou Prov.; Clade B was O. tianmuii occurring from Huangshan Mountains in Anhui Prov. and Tianmu Mountains in Zhejiang Prov.; Clade C was O. schmackeri sensu stricto occupying a more narrow distribution area in Funiu Mountains in Henan and northern Hunan provinces and Daba Mountains in Hubei Prov.; Clade D was proposed as a cryptic species from mountains in Jiangxi Prov.; Clade E was the second cryptic species from mountains of northwestern Guizhou Prov.; Clade F was the third cryptic species from Funiu Mountains in Henan Prov. and being sympatric with Clade C; and Clade G was the fourth cryptic species from Daba Mountains in Hubei Prov. and also being sympatric with O. schmackeri sensu stricto (Fig. 1). Also, Zhu (2016) based on 12S rRNA and 16S rRNA genes showed a broadly similar phylogeographic framework through a more comprehensive sampling with 78 populations, and supplied several different implications: Clade D in Li et al. (2015) should not be recognized as a cryptic species but still be classified as O. schmackeri, and thus O. schmackeri sensu stricto was in fact distributed in western Henan, southeastern Shanan’xi, eastern to central Chongqing, northwestern and eastern Hubei, northwestern Hunan, northeastern Guizhou and Jiangxi provinces in China; O. hejiangensis occupied a large range around Sichuan Basin even in western Henan and eastern Shananxi provinces; one cryptic species (Odorrana sp1 defined by them) had a large range across central Guizhou and Guangxi provinces; and another cryptic species (Odorrana sp2 defined by them) occupied a large range around the southern part of Sichuan Basin (Fig. 1). However, because of different samplings and use of different genes in the two studies especially Li et al. (2015) having no morphological data and no releasing of sequences in Zhu (2016), there were still several uncertain points: (1) whether Clade E in Li et al. (2015) containing only two populations (Jinsha Co. and Suiyang Co. in Guizhou Prov.) belonged to Odorrana sp1 in Zhu (2016); (2) whether Clade F (the third cryptic species) from Funiu Mountains in Henan Prov. in Li et al. (2015) belonged to O. hejiangensis; (3) whether Clade G from Daba Mountains in Li et al. (2015) belonged to Odorrana sp2 in Zhu (2016). Anyway, several cryptic species in the species complex have been indicated. To better understand diversification of the species complex, it is necessary and urgent to make more investigations (e.g., distribution range, morphology, molecular phylogenetics, ecology, and tadpoles) on the new taxa.

Figure 1 Sampling localities in this study.

Localities 1–8 were all in China. (1) Jinsha County (Co.), Guizhou Province (Prov.); (2) Zheng’an Co., Guizhou Prov.; (3) Meitan Co., Guizhou Prov.; (4) Leishan Co., Guizhou Prov.; (5) Jiangkou Co., Guizhou Prov.; (6) Sangzhi Co., Hunan Prov.; (7) Changyang Co., Hubei Prov.; (8) Ruyuan Co., Guangdong Prov. According to literatures and results in this study, the distributional range of each related species was surrounded by dotted lines on the map. Odorrana sp1 and Odorrana sp2 were proposed in Zhu (2016); Odorrana sp3 was corresponding to the third cryptic species (Clade F) proposed in Li et al. (2015). Different species were denoted as different colors and shapes.

From 2013 to 2017, we carried out a series of biodiversity surveys in Guizhou Prov. of China and collected many specimens of Odorrana superficially resembling O. schmackeri. As noted, some of them were collected from the population in the Lengshuihe Reserve in Jinsha Co., Guizhou Prov., China (Fig. 1) where the samples of Clade E in Li et al. (2015) indicated as a cryptic species was collected. To distinguish these specimens, we conducted phylogenetic analyses based on mitochondrial DNA and nuclear DNA and morphological comparisons. All analyses consistently supported our specimens from Guizhou Prov. as a new taxon. Hence, we describe it herein as a new species.

Methods

Sampling

Frogs were collected on September 17 and 21, 2013 and August 3, 4, and 8, 2015, and tadpoles were collected on October 10, 2017. After taking photographs, they were euthanized using isoflurane, and then the specimens were fixed in 10% buffered formalin. Tissue samples were taken and preserved separately in 95% ethanol prior to fixation. Specimens collected in this work were deposited in Chengdu Institute of Biology, Chinese Academy of Sciences (CIB, CAS).

A total of 25 adult specimens of the new taxon including 16 females and nine males were collected from the mountain streams of three localities across Guizhou Province of China (Fig. 1; for voucher information see Table S1). For comparisons, some specimens of the related species were also collected in this study, including 19 O. schmackeri specimens from two localities, 23 O. huanggangensis specimens from three localities, three O. yizhangensis specimens from two localities and two O. lungshengensis specimens from one locality (Fig. 1; for voucher information see Table S1).

A total of 10 tadpoles with almost identical morphology were collected from the same place in the stream where the new taxon was found in the Lengshuihe Nature Reserve of Jinsha Co., Guizhou Prov., China. They were identified as the new taxon because they were almost identical in morphology and one representative of them was genetically close to the adult specimens of the new taxon (see Results). Stages of tadpoles were identified following Gosner (1960).

Collection of molecular data

A total of 18 molecular samples were collected in this study: five containing four adults and one tadpole of the new taxon, five of O. schmackeri including two topotypes, three of O. huanggangensis, three of O. yizhangensis including one topotype, and two of O. lungshengensis (for voucher information see Table S2).

Total DNA was extracted using a standard phenol–chloroform extraction protocol (Sambrook, Fritsch & Maniatis, 1989). The mitochondrial 12S rRNA, 16S rRNA, and ND2 genes and two nuclear protein-coding genes (DOLK and KCNF genes) were amplified and sequenced from our samples. Primer sequences were acquired from literatures for 12S rRNA (Kocher et al., 1989), 16S rRNA (Simon et al., 1994), ND2 (Li et al., 2015), DOLK (Shen et al., 2013), and KCNF (Shen et al., 2013) genes (primer sequences were presented in Table S3). PCR amplification reactions for mitochondrial genes were performed in a 30 μL volume reaction with the following cycling conditions: an initial denaturing step at 95 °C for 4 min; 36 cycles of denaturing at 95 °C for 40 s, annealing at 55 °C (for 12S and 16S)/47–57 °C (for ND2) for 40 s and extending at 72 °C for 70 s, and a final extending step of 72 °C for 10 min. Amplifications of nuclear genes were according to Shen et al. (2013). PCR products were purified with spin columns and then were sequenced with both forward and reverse primers same as used in PCR. Sequencing was conducted using an ABI3730 automated DNA sequencer in Shanghai DNA BioTechnologies Co., Ltd. (Shanghai, China). All new sequences were deposited in GenBank (for GenBank accession numbers see Table S2).

For phylogenetic analyses, we downloaded 12S and 16S gene sequences from GenBank for all those related species especially for their topotypes for which comparable sequences were available (for GenBank Accession numbers see Table S2) based on the previous studies (Chen et al., 2013; Li et al., 2015). As noted, ND2 gene sequences for most Odorrana species were not sequenced up to now, impeding us in examining comprehensive relationships of the genus using this gene. To further understand the divergence between the new taxon and its related species, ND2 sequences for all haplotypes of the O. schmackeri species complex in Li et al. (2015) were downloaded (for GenBank accession number see Table S2).

Phylogenetic analyses and genetic distance estimation

Sequences were assembled and aligned using the ClustalW module in BioEdit v. 7.0.9.0 (Hall, 1999) with default settings. The datasets were checked by eye and revised manually if necessary. To avoid bias in alignments, GBLOCKS v. 0.91.b (Castresana, 2000) with default settings was used to extract regions of defined sequence conservation from the length-variable 12S and 16S fragments. No-sequenced fragments were treated as missing data. Finally, for phylogenetic analyses of mitochondrial DNA, two datasets were obtained, that is, three-genes concatenated dataset with 12S (for 61 samples) + 16S (for 61 samples) + ND2 (for 16 samples) and ND2 gene alone dataset (for 112 samples).

Based on the three-genes concatenated dataset, phylogenetic analyses were conducted using maximum likelihood (ML) and Bayesian Inference (BI) methods, implemented in PhyML v. 3.0 (Guindon et al., 2010) and MrBayes v. 3.12 (Ronquist & Huelsenbeck, 2003), respectively. One Rana chensinensis was chosen as outgroup according to the previous studies (Pyron & Wiens, 2011; Chen et al., 2013). To avoid under- or over-parameterization (Lemmon & Moriarty, 2004; McGuire et al., 2007), for the phylogenetic analyses, the best partition scheme and the best evolutionary model for each partition were chosen using PARTITIONFINDER v. 1.1.1 (Lanfear et al., 2012). In this analysis, 12S, 16S genes and each codon position of ND2 gene were defined and Bayesian Inference Criteria was used. As a result, the analysis suggested that the best partition scheme is 12S gene/16S gene/each codon position of ND2 gene, and selected TrN + I + G model as the best model for 12S, 16S and the second codon position of ND2 gene and GTR + I + G model as the best model for the other two codon position of ND2 gene. For the ML tree, branch supports were drawn from 10,000 nonparametric bootstrap replicates. In BI analyses, the parameters for each partition were unlinked, and branch lengths were allowed to vary proportionately across partitions. Two runs each with four Markov chains were simultaneously run for 50 million generations with sampling every 1,000 generations. The first 25% trees were removed as the “burn-in” stage followed by calculations of Bayesian posterior probabilities and the 50% majority-rule consensus of the post burn-in trees sampled at stationarity.

To further visualize the degree of genetic splits among the new taxon and its related species especially members of the O. schmackeri species complex recognized by Li et al. (2015), a phylogenetic network using the maximum parsimony method in Splittree v. 4.11.3 (Huson & Bryant, 2006) was constructed based on the ND2 gene sequence dataset. The supports of Splittree lineages were evaluated by 1,000 bootstrap replicates.

In addition, to access the genetic divergence between the new taxon and its related species on nuclear DNA, haplotype networks for DOLK and KCNF gene datasets were constructed, respectively, using the maximum parsimony method in TCS v. 1.21 (Clement, Posada & Crandall, 2000).

Finally, pairwise uncorrected p-distance on the 16S rRNA gene were estimated using MEGA v. 6.06 (Tamura et al., 2011) to evaluate the genetic divergence between Odorrana species.

Morphological comparisons

A total of 67 adult specimens including nine males and 16 females of the new taxon, 15 males and four females of O. schmackeri and 13 males and ten females of O. huanggangensis were measured (for voucher information see Table S1). Ten tadpoles of the new taxon were measured (for voucher information see Table S4). The terminology and methods followed Fei et al. (2009). Measurements were taken with a dial caliper to the nearest 0.1 mm. Twenty one morphometric characters of adult specimens were measured: SVL (distance from the tip of the snout to the posterior edge of the vent), head length (HDL; distance from the tip of the snout to the articulation of jaw), maximum head width (HDW; greatest width between the left and right articulations of jaw), snout length (SL; distance from the tip of the snout to the anterior corner of the eye), eye diameter (ED; distance from the anterior corner to the posterior corner of the eye), interorbital distance (IOD; minimum distance between the inner edges of the upper eyelids), internasal distance (IND; minimum distance between the inner margins of the external nares), nasal to eye distance (NED; distance between the nasal and the anterior corner of the eye), nasal to snout distance (NSD; distance between the nasal the posterior edge of the vent), IFE (distance between anterior corner of eyes), IAE (distance between posterior corner of eyes), maximal tympanum diameter (TYD), length of lower arm and hand (LAL; distance from the elbow to the distal end of the Finger IV), lower arm width (LW; maximum width of the lower arm), thigh length (THL; distance from vent to knee), tibia length (TL; distance from knee to tarsus), maximal tibia width (TW), length of foot and tarsus (TFL; distance from the tibiotarsal articulation to the distal end of the Toe IV), foot length (FL; distance from tarsus to the tip of fourth toe), finger disk width (FDW; width at the widest part of the disk of finger III), distal phalanx width (DPW; maximal width of distal phalanx of finger III). A total of 10 morphometric characters of larvae were measured: total length (TOL), SVL, maximum body height (BH), maximum body width (BW), SL (distance from the anterior corner of the eye to the tip of the snout), snout to spiraculum (SS; distance from spiraculum to the tip of the snout), mouth width (MW; distance between two corners of mouth), maximum width of tail base (TBW), tail length (TL; distance from base of vent to the tip of tail), tail height (TH; maximum height between upper and lower edges of tail).

In order to reduce the impact of allometry, the correct value from the ratio of each measurement to SVL was calculated and then log-transformed for the following morphometric analyses. One-way analysis of variance (ANOVA) was used to test the significance of differences on morphometric characters between males and females and between different species. The significance level was set at 0.05. To show the spatial distribution of different species on the morphometric characters, principal component analyses (PCA) were performed. These analyses were carried out in the R (R Development Core Team, 2008).

The new species was also compared with all other Odorrana species on morphology. Comparative morphological data were obtained from literatures for O. absita (Stuart & Chan-ard, 2005), O. amamiensis (Matsui, 1994), O. andersonii (Boulenger, 1882), O. anlungensis (Hu, Zhao & Liu, 1973), O. arunachalensis (Saikia, Sinha & Kharkongor, 2017), O. aureola (Stuart et al., 2006), O. bacboensis (Bain et al., 2003), O. banaorum (Bain et al., 2003), O. bolavensis (Stuart & Bain, 2005), O. cangyuanensis (Yang, 2008), O. chapaensis (Bourret, 1937), O. chloronota (Günther, 1876), O. exiliversabilis (Li et al., 2001 in Fei, Ye & Li, 2001b), O. fengkaiensis (Wang et al., 2015), O. geminata (Bain et al., 2009), O. gigatympana (Orlov, Natalia & Cuc, 2006), O. grahami (Boulenger, 1917), O. graminea (Boulenger, 1900 “1899”), O. hainanensis (Fei, Ye & Li, 2001a), O. hejiangensis (Deng & Yu, 1992), O. hosii (Boulenger, 1891), O. huanggangensis (Chen, Zhou & Zheng, 2010a), O. heatwolei (Stuart & Bain, 2005), O. hmongorum (Bain et al., 2003), O. indeprensa (Bain & Stuart, 2005), O. ishikawae (Stejneger, 1901), O. jingdongensis (Fei, Ye & Li, 2001a), O. junlianensis (Fei & Ye, 2001 in Ye & Fei, 2001), O. khalam (Stuart, Orlov & Chan-ard, 2005), O. kuangwuensis (Liu & Hu, 1966 in Hu, Zhao & Liu, 1966), O. leporipes (Werner, 1930), O. lipuensis (Mo et al., 2015), O livida (Blyth, 1856 “1855”), O. macrotympana (Yang, 2008), O. margaretae (Liu, 1950), O. mawphlangensis (Pillai & Chanda, 1977), O. monjerai (Matsui & Jaafar, 2006), O. morafkai (Bain et al., 2003), O. mutschmanni (Pham et al., 2016), O. nanjiangensis (Fei, Ye & Xie, 2007), O. narina (Stejneger, 1901), O. nasica (Boulenger, 1903), O. nasuta (Li et al., 2001 in Fei, Ye & Li, 2001b), O. orba (Stuart & Bain, 2005), O. rotodora (Yang & Rao, 2008 in Yang, 2008), O. schmackeri (Boettger, 1892), O. sinica (Ahl, 1927 “1925”), O. splendida (Kuramoto et al., 2011), O. supranarina (Matsui, 1994), O. swinhoana (Boulenger, 1903), O. tianmuii (Chen, Zhou & Zheng, 2010b), O. tiannanensis (Yang & Li, 1980), O. tormota (Wu, 1977), O. trankieni (Orlov, Ngat & Cuc, 2003), O. utsunomiyaorum (Matsui, 1994), O. versabilis (Liu & Hu, 1962), O. wuchuanensis (Xu, 1983 in Wu et al., 1983), O. yentuensis (Tran, Orlov & Nguyen, 2008), O. yizhangensis (Fei, Ye & Jiang, 2007), and O. zhaoi (Li, Lu & Rao, 2008).

Skull scanning

Skulls of two male specimens (voucher number: CIBjs20150803001, CIBjs20150803002) and three female specimens (voucher number: CIBjs20150804001, CIBGYU20130917003, CIBGYU20130917001) of the new taxon were scanned in the high-resolution X-ray scanner (Quantum GX micro-CT Imaging System; PerkinElmer®, Boston, MA, USA). The specimens were scanned along the coronal axis at an image resolution of 2,000 × 2,000. Segmentation and three-dimensional reconstruction of the CT images were made using VG57 Studio Max 2.2 (Volume Graphics, Heidelberg, Germany).

The Animal Care and Use Committee of Chengdu Institute of Biology, CAS provided full approval for this purely observational research (Number: CIB2013041102). Field work was approved by the Management Office of the Lengshuihe Nature Reserve (project number: LSH-201304003).

The electronic version of this article in portable document format will represent a published work according to the International Commission on Zoological Nomenclature (ICZN), and hence the new names contained in the electronic version are effectively published under that Code from the electronic edition alone. This published work and the nomenclatural acts it contains have been registered in ZooBank, the online registration system for the ICZN. The ZooBank LSIDs (Life Science Identifiers) can be resolved and the associated information viewed through any standard web browser by appending the LSID to the prefix http://zoobank.org/. The LSID for this publication is: urn:lsid:zoobank.org:pub:E98B65CB-E8E3-4412-9613-D9DD32A77B99. The online version of this work is archived and available from the following digital repositories: PeerJ, PubMed Central, and CLOCKSS.

Results

Phylogenetic analyses and genetic divergence

Aligned sequence matrix of 12S + 16S, ND2, DOLK, and KCNF genes contained 1,835, 768, 645, and 750 bp, respectively. ML and BI analyses based on the 12S + 16S + ND2 matrix resulted in essentially identical topologies with high node supporting values (Fig. 2A). All samples of the new taxon occurring from Guizhou Prov. were strongly clustered into a monophyly, which was placed into the genus Odorrana and sister to the O. schmackeri clade. ND2 splitstree also strongly supported the splits between the new taxon and its related species (Fig. 2B). As noted, Odorrana sp2 simultaneously revealed by Li et al. (2015) and Zhu (2016) and Odorrana sp3 revealed by Li et al. (2015) also occupied an independent lineage, respectively. Only one haplotype was found for all samples of the new taxon, either in KCNF gene or in DOLK gene, and there was no common haplotype between the new species and its related species (Fig. 3).

Figure 2 Phylogenetic relationships of Odorrana kweichowensis sp. nov. and its congeners.

(A) Maximum likelihood (ML) tree reconstructed based on the three genes (12S + 16S + ND2) concatenated dataset. (B) Phylogenetic network of O. kweichowensis sp. nov. and its related species reconstructed by the software Splittree based on ND2 gene sequences. In ML tree, bootstrap supports (BS) from ML analyses/Bayesian posterior probabilities (BPP) from BI analyses were noted beside nodes. In Splittree network, numbers on branches are BS. Odorrana sp2 was proposed in Zhu (2016); Odorrana sp3 corresponded to the third cryptic species (Clade F) in Li et al. (2015). Different related species of Odorrana kweichowensis sp. nov. were denoted as different colors.

Figure 3 Haplotype networks of O. kweichowensis sp. nov. and its related species constructed based on the nuclear gene sequences.

(A) DOLK gene. (B) KCNF gene. Different species were denoted as different colors.

The genetic distances on 16S gene between the new taxon and its sister species O. schmackeri was mean 0.027 (range 0.026–0.028; Table S5), much higher than the intraspecific genetic distance within each of the new taxon, O. schmackeri, O. huanggangensis, O. yizhangensis, and O. lungshengensis (all intraspecific genetic distance <0.002). More significantly, it was higher or even much higher than the interspecific genetic distance between many sister species, for example, that between O. huanggangensis and O. tianmuii (0.014), O. hainanensis and O. fengkaiensis (0.013), O. nasuta and O. versabilis (0.013), O. versabilis and O. exiliversabilis (0.017), O. morafkai and O. banaorum (0.013), and O. grahami and O. jingdongensis (0.017; Table S5).

Morphological comparisons

The results of one-way ANOVA showed that in the new taxon, the males was significantly different from the females on SVL and the ratios of ED, IND, IFE, LAL, TYD, LW, and TW to SVL (all p-values < 0.05; Table 1). Therefore, morphometric analyses were conducted on males and females, respectively. In PCA for males, the total variation of the first two principal components was 42.23%, and in PCA for females, it is 37.37%. In both males (Fig. 4A) and females (Fig. 4B), on the two-dimensional plots of PC1 vs. PC2, the new taxon could be almost separated from O. schmackeri. The results of one-way ANOVA indicated that either in males or females, the new taxon was significantly different from O. schmackeri and O. huanggangensis on many morphometric characters (all p-values < 0.05; Table 1). More detailed descriptions of results from morphological comparisons between the new taxon and its congeners were presented in the following sections for describing the new species.

Table 1 The results of one-way ANOVA with p-values for morphometric comparisons between Odorrana kweichowensis sp. nov., O. schmackeri, and O. huanggangensis.

	Males of OK vs. Females of OK	In males	In females	
OK vs. OS	OK vs. OH	OK vs. OS	OK vs. OH	
SVL	0.0001***	0.3160	0.2370	0.0810	0.4520	
HDL	0.5560	0.0030**	0.5740	0.0900	0.0540	
HDW	0.6240	0.2100	0.7650	0.0560	0.9000	
SL	0.9840	0.0130*	0.1130	0.0740	0.9860	
ED	0.0001***	0.4730	0.2510	0.1560	0.9130	
IOD	0.1470	0.0910	0.2370	0.4520	0.2880	
IND	0.0210*	0.5770	0.0330*	0.0290*	0.8340	
NED	0.1470	0.2790	0.8480	0.4940	0.8860	
NSD	0.8830	0.0980	0.0430*	0.3060	0.0310*	
IFE	0.0001***	0.2140	0.3870	0.0740	0.6930	
IAE	0.0001***	0.0490*	0.0310*	0.2320	0.2060	
TYD	0.0001***	0.1960	0.2980	0.9260	0.1040	
LAL	0.3340	0.0950	0.0310*	0.0320*	0.0000***	
LW	0.0040**	0.1090	0.1390	0.3510	0.6420	
THL	0.7560	0.0110*	0.0240*	0.0270*	0.0000*	
TL	0.0920	0.0170*	0.1970	0.0420*	0.0040**	
TW	0.0220*	0.0840	0.1430	0.1500	0.0160*	
TFL	0.9930	0.0540	0.0290*	0.0500	0.0390	
FL	0.2690	0.0170*	0.2410	0.0270	0.0050**	
FDW	0.3400	0.0840	0.5660	0.9480	0.5520	
DPW	0.5490	0.1040	0.6290	0.4520	0.6510	
Notes:

OK, O. kweichowensis sp. nov.; OS, O. schmackeri; OH, O. huanggangensis.

Significance level:

* p < 0.05;

** p < 0.01;

*** p < 0.001.

Abbreviations for the morphometric characters refer to Methods section.

Figure 4 Plots of principal component analyses of Odorrana kweichowensis sp. nov., O. schmackeri, and O. huanggangnesis.

(A) Males. (B) Females. PC1, the first principal component; PC2, the second principal component. Different species were denoted as different colors and shapes.

At all, molecular and morphological results supported that our specimens from Guizhou Prov. of China was a new taxon. It is described as a new species in the following sections: Odorrana kweichowensis sp. nov.

urn:lsid:zoobank.org:act:95123118-30D2-42B5-B5DB-AB7C32B33F97

Holotype

CIBjs20150803002, adult male (Figs. 5A, 5B, 6A and 6B), collected by S. Z. Li in the Lengshuihe Nature Reserve (27.47361°N, 106.00139°E; elevation 754 m a.s.l.), Jinsha Co., Guizhou Prov., China.

Figure 5 Comparisons of the holotype (voucher number: CIBjs20150803002) of Odorrana kweichowensis sp. nov. and one male specimen (voucher number: CIBsz2012062003) of O. schmackeri.

(A) and (B) Dorsal view and ventral view of specimen CIBjs20150803002, respectively. (C) and (D) Dorsal view and ventral view of specimen CIBsz2012062003, respectively. (E) Ventral view of hand of specimens CIBjs20150803002. (F) Ventral view of hand of specimen CIBsz2012062003. (G) Ventral view of foot of specimen CIBjs20150803002. (H) Ventral view of foot of specimen CIBsz2012062003. Photographs by S. Z. Li.

Figure 6 Living Odorrana kweichowensis sp. nov. from its type locality, Lengshihe Nature Reserve in Jinsha County, Guizhou Province, China.

(A & B) Dorsolateral view and ventral view of an adult male (voucher number: CIBjs20150803002), respectively. (C & D) Dorsolateral view and ventral view of an adult female (voucher number: CIBjs20150803006), respectively. Photographs by S. Z. Li.

Paratypes

A total of 24 specimens (nine adult males and 15 adult females), 14 specimens collected by S. Z. Li from the Lengshuihe Nature Reserve in Jinsha Co., Guizhou Prov., China. Six males: CIBjs20150803001, CIBjs20150803003, CIBjs20150803004, CIBjs20150803005 collected on August 3, 2015 and CIBjs20150804002 on August 4, 2015; eight females: CIBjs20150803006, CIBjs20150803007, CIBjs20150803008 collected on August 3, 2015, CIBjs20150804001, CIBjs20150804003, CIBjs20150804004, CIBjs20150804005 on August 4, 2015 and CIBjs20150808020 on August 8, 2015. Seven specimens collected by S. Z. Li and J. C. Lv from Meitan Co. of Guizhou Prov., China. Three males: CIBGYU20130917004, CIBGYU20130917005, and CIBGYU20130917007 collected by J. C. Lv on September 17, 2013; four females: CIBGYU20130917001, CIBGYU20130917002, and CIBGYU20130917003 by S. Z. Li on September 17, 2013, CIBGYU20130917006 by J. C. Lv on September 17, 2013. Three females: GYU20130921001, CIBGYU20130921002, and CIBGYU20130921003 collected by S. Z. Li on September 21, 2013. A total of 10 tadpoles (CIBjs20171014001–CIBjs20171014010) collected by S. Z. Li on October 10, 2017.

Diagnosis

Odorrana kweichowensis sp. nov. is assigned to genus Odorrana based upon molecular phylogenetic analyses and the following morphological characters: dorsum is green; tips of digits dilated, tapering, disks with circum-marginal grooves, and vertical diameter longer than horizontal diameter in the disks; supernumerary tubercle below the base of fingers III and IV; feet fully webbed to disks, without tarsal fold; the first finger thick and nuptial pad distinct.

Odorrana kweichowensis sp. nov. could be distinguished from its congeners by a combination of the following characters: (1) having smaller body size in males (SVL <43.3 vs. SVL >48 mm in many other species); (2) head longer than wider; (3) dorsolateral folds absent; (4) tympanum of males large and distinct, tympanum diameter in males twice as long as width of distal phalanx of finger III; (5) two metacarpal tubercles; (6) relative finger lengths: II < I < IV < III; (7) tibiotarsal articulation reaching to the level between eye to nostril when leg stretched forward; (8) disks on digits with circum-marginal grooves; (9) toes fully webbed to disks; (10) the first subarticular tubercle on fingers weak; (11) having white pectoral spinules, paired subgular vocal sacs located at corners of throat, light yellow nuptial pad on the first finger in males.

Description of holotype

Head longer than wide (HDL/HDW = 1.08); top of head flat; snout obtusely rounded in dorsal view (SL/HDL = 0.41), rounded in profile, projecting beyond lower jaw; eye large and convex, ED 0.83 times of SL; IND = 5.1 mm, larger than IOD = 3.2 mm; NED = 3.4 mm larger than NSD = 2.9 mm; tympanum circular, large and distinct, twice as long as width of distal phalanx of finger III (TYD/FDW = 2.84); vomerine teeth on well-developed ridges; tongue deeply notched posteriorly; paired gular pouches at corners of throat. Forelimbs moderately robust (LW/SVL = 0.11); lower arm and hand beyond a half of body length (LAL/SVL = 0.53); the first finger slightly longer than the second; finger tips on I–IV dilated to wide, tapering, disks with circum-marginal grooves; nuptial pad on the inner of first finger from base to subarticular tubercles; subarticular tubercles relatively prominent; inner metacarpal tubercle oblong and outer metacarpal tubercle indistinct; no finger webbing.

Hindlimbs long; tibiotarsal articulation reaching to the level between eye to nostril when hindlimb adpressed along the side of body; heels overlapping when hindlimbs positioned at right angle to body; singular longer than thigh length; FL 0.85 times SVL; toes slender, relative toe lengths: I < II < III < V < IV; toes tapering dilated, disks with circum-marginal grooves; feet fully webbed to disks; web becoming narrower and continuing to the disks as lateral fringe on toes II, III, and IV; outer metatarsal tubercle absent; inner metatarsal tubercle present.

Skin shagreened, a number of pustules scattered on dorsum of trunk and flanks; several irregular tubercles scattered on flanks; dorsolateral folds absent; two large glands on the place between below tympanum and above arm; white spinules on throat, chest, fore abdomen, inner side of forearms and upper and lower lips.

Coloration in life

In life, dorsum with bright green network; large black spots in the center on dorsum, continuing onto dorsal portion of legs to form dark counter-point to bright green irregular bands; flank light yellow with several black spots; tympanum brown–black; upper and lower lip with vertical black bars; arms brown with black transverse bands, thighs with five brown bands and tibias with six (Fig. 6).

Color in preservative

On dorsum color fades to dark olive with dark brown blotches (Fig. 5A), upper and lower lips marbled brown–black on dirty white; venter variable from white to light yellow; underside of limbs yellowish with white (Fig. 5B); pinkish in the inner side of forearm and thighs; there is a white spot between front corner of eyes; nuptial pad fades to white (Fig. 5).

Variation

Basic statistics for measurements were presented in Table S6. All male specimens were similar in morphology and color pattern, but different from females. SVL in females approximately 1.8 times that in males (SVL mean 73.6 mm, range 62.4–81.1 mm in females, SVL mean 41.0 mm, range 36.2–43.3 mm in males); the ratios of ED, IND, IFE, LAL, TYD, and LW to SVL of males are significantly higher than that of females, but the ratios of TW to SVL of males is lower than that of females (Table 1). In some adult females, the black transverse bands on limbs are not obvious (Fig. 6C). In some adult females, chest, chin and ventral surface of limbs scattered with continued light-brown spots (Fig. 7A). Some adult females have a black cloud stripe with irregular borders on the ventral side (Fig. 7B), but some don’t have this trait (Fig. 6D). In some adult females, dorsum is uniform brown with little green impression (Fig. 7C) and some adult females have dorsum of uniform brown lacking a green impression (Fig. 7D).

Figure 7 Color variations in Odorrana kweichowensis sp. Nov.

(A) and (B) Dorsolateral view and ventral view of an adult female from Jinsha County, Guizhou Province, China, respectively. (C) Dorsolateral view of a female from Meitan County, Guizhou Province, China. (D) Dorsolateral view of a female from Zheng’an County, Guizhou Province, China. Photographs by S. Z. Li.

Skull description

The skull morphology of the five scanned specimens were almost identical, and thus, only one representative (voucher number: CIBjs20150803001) was presented (Fig. 8) and described as following: skull flat, maxillary teeth well developed, vomerine teeth present; mandible without teeth; nasals large, widely separated from each other, nasals disconnected with the sphenethmoid; sphenethmoid tubular, forms the anterolateral walls of the braincase; frontoparietal roof the braincase and wider posteriorly than anteriorly; prootic large and connect with the exoccipital; a pair of exoccipitals situated the end of the brain; palatines arcuate and long, behind the prevomers; a pair of prevomers obliquely lie anterior to the palatines, vomerine teeth on the prevomers indistinct; parasphenoid in sword shape, supports the braincase ventrally, connects with palatines; a pair of squamosals on the dorsolateral side of the prootic, each squamosal consists of three rami: the zygomatic ramus, the otic ramus, and the ventral ramus; zygomatic ramus pointing to the orbit, otic ramus shorted, ventral ramus outboard and covered the posterior of pterygoid; a pair of pterygoids, outside of the ventral surface of the squamosal, anterior ramus is the longest, center is inward and leading edge extends to the orbit, medial ramus shorted and attached to the anterior lateral part of the prootic, posterior ramus edge-on and extend to maxillary; a pair of columellaes situated ventral to the crista parotica (Fig. 8).

Figure 8 Skull of Odorrana kweichowensis sp. Nov.

(A) Dorsal view. (B) Ventral view. (1) Maxillary; (2) nasal; (3) frontoparietal; (4) pterygoid; (5) squamosal; (6) exoccipital; (7) prootic; (8) maxillary teeth; (9) prevomer; (10) palatine; (11) mandible; (12) sphenethmoid; (13) parasphenoid; (14) columella. Drawings by S. Z. Li.

Tadpole description

Body and tail yellowish-brown (Fig. 9A), at Gosner’s stage 28–29, TOL 31.6–36.2 mm, SVL 12.1–14.5 mm, other measurements of tadpoles shown in Table S4. Tail 1.5–1.8 times as long as body; TH 25–30% of TL in the 28th stage and 23–28.5% in the 29th stage; BW 45.8% of SVL in the 28th stage and 46.8% in the 29th stage; BH 31.4% of SVL in the 28th stages and 34.9% in the 29th stage; tail fins lightly colored, tail muscles with small black spots; tail depth greater than body depth, dorsal fin arising behind the origin of the tail; maximum depth near mid-length, tip of tail blunt; nostril near snout, eyes positioned dorsally (Fig. 9A); spiracle on the middle left of body (Fig. 9B); keratodont formula I: 3–3/III: 1–1 (Fig. 9C); both upper and lower lips with labial papillae (Fig. 9C); some additional tubercles at the angles of the mouth, usually with small keratodonts (Fig. 9C).

Figure 9 Tadpole of Odorrana kweichowensis sp. Nov.

(A) Dorsolateral view of specimen CIBJS20171014001 in life. (B) Dorsal view of specimen CIBJS20171014001 in preservative. (C) Structure of the mouth of specimen CIBJS20171014001. (1) Spiracle; (2) upper keratodonts; (3) lower keratodonts; (4) labial papillae on upper lips; (5) labial papillae on lower lips; (6) additional tubercles at the angles of mouth. Photographs by S. Z. Li.

Sexual dimorphism

Snout-vent length in females approximately 1.8 times that in males (SVL mean 73.6 mm, range 62.4–81.1 mm in females; SVL mean 41.0 mm, range 36.2–43.3 mm in males); paired subgular vocal sacs located at corners of throat, pinkish lineae musculinae on dorsal side, nuptial pad light yellow on the inner of first finger from base to subarticular tubercles in males (Figs. 5 and 6).

Comparisons

Odorrana kweichowensis sp. nov. differs from O. amamiensis, O. andersonii, O. bacboensis, O. cangyuanensis, O. chapaensis, O. geminata, O. graham, O. hmongorum, O. ishikawae, O. jingdongensis, O. junlianensis, O. kuangwuensis, O. macrotympana, O. margaretae, O. mawphlangensis, O. mutschmanni, and O. wuchuanensis, by having smaller body size (SVL <43.30 mm in males in the new species vs. SVL >48 mm in males in other species).

Odorrana kweichowensis sp. nov. differs from O. absita, O. aureola, O. banaorum, O. bolavensis, O. exiliversabilis, O. gigatympana, O. graminea, O. hosii, O. heatwolei, O. indeprensa, O. khalam, O. leporipes, O. livida, O. monjerai, O. narina, O. nasica, O. nasuta, O. orba, O. supranarina, O. tormota, O. trankieni, O. utsunomiyaorum, O. versabilis, O. yentuensis, and O. zhaoi, by lacking dorsolateral folds (vs. present in other species).

Odorrana kweichowensis sp. nov. differs from O. fengkaiensis by tibiotarsal articulation reaching the level between eye to nostril when leg stretched forward (vs. reaching the level below nostril in the latter); from O. rotodoar by tibiotarsal articulation reaching the level between eye to nostril when leg stretched forward (vs. reaching the level beyond eye in the latter); from O. lipuensis (vocal sacs absence in males) and O. hainanensis (paired internal vocal sacs in males) by paired external vocal sacs located at corners of throat; from O. chloronota by having head longer than wide (vs. head wider than long in the latter); from O. morafkai and O. sinica by having white pectoral spinules in mature males (vs. absence in the latter); from O. arunachalensis, by the relative finger lengths I < II < IV < III (vs. II < I < IV < III in the latter).

Within the O. schmackeri group (O. anlungensis, O. hejiangensis, O. lungshengensis, O. nanjiangensis, O. swinhoana, O. tianmuii, O. tiannanensis, O. yizhangensis, O. huanggangensis, and O. schmackeri), O. kweichowensis sp. nov. differs from O. anlungensis by having white pectoral spinules in mature males (vs. absence in the latter), tibiotarsal articulation reaching the level between eye to nostril when leg stretched forward (vs. reaching the nostril in the latter), web becoming narrower and continuing to the disks of toe IV (vs. web continuing to the subarticular tubercles of toe IV in the latter); from O. hejiangensis by the disks of fingers with circum-marginal grooves (vs. without circum-marginal grooves on finger I in the latter), two metacarpal tubercles (vs. three metacarpal tubercles in the latter); from O. lungshengensis by the body size of males with maximum SVL 43.3 mm (vs. SVL 50.0–59.6 mm of males in the latter), tympanum large and distinct, and beyond twice as long as width of distal phalanx of finger III (vs. tympanum as long as width of distal phalanx of finger III in the latter), two metacarpal tubercles (vs. without outer metacarpal tubercle in the latter), tibiotarsal articulation reaching the level between eye to nostril when leg stretched forward (vs. reaching the tip of snout in the latter); from O. nanjiangensis by the body size in males with maximum SVL 43.3 mm (vs. SVL 59.5–66.8 mm in males in the latter), two metacarpal tubercles (vs. without outer metacarpal tubercle in the latter), tibiotarsal articulation reaching the level between eye to nostril when leg stretched forward (vs. reaching the tip of snout in the latter); from O. swinhoana by the body size in males with maximum SVL 43.3 mm (vs. SVL 47.7–71.6 mm in males in the latter), tibiotarsal articulation reaching the level between eye to nostril when leg stretched forward (vs. reaching the tip of snout); from O. tianmuii by the relative finger lengths II < I < IV < III (vs. I < II < IV < III in the latter), two metacarpal tubercles (vs. three outer metacarpal tubercles in the latter); O. tiannanensis by the body size in males with maximum SVL 43.3 mm (vs. SVL 52.5–53.5 mm in males in the latter), tibiotarsal articulation reaching the level between eye to nostril when leg stretched forward (vs. reaching the level beyond the tip of snout in the latter); O. yizhangensis by the body size in males with maximum SVL 43.3 mm (vs. SVL 47.3–54.0 mm in males in the latter), two metacarpal tubercles (vs. without outer metacarpal tubercle), tibiotarsal articulation reaching the level between eye to nostril when leg stretched forward (vs. reaching the tip of snout in the latter); from O. huanggangensis by the relative finger lengths II < I < IV < III (vs. I < II < IV < III in the latter), tibiotarsal articulation reaching the level between eye to nostril when leg stretched forward (vs. reaching the nostril in the latter); from O. schmackeri, by the first subarticular tubercle on fingers and supernumerary tubercles being weak (vs. being outstanding in the latter; Figs. 5E and 5F), and having two outer metacarpal tubercles (vs. outer metacarpal tubercle being indistinct in the latter; Figs. 5H and 5G).

Odorrana kweichowensis sp. nov. differs from O. schmackeri by having significantly lower ratios of HDL and SL to SVL in males, having significantly higher ratios of THL, TL, and FL to SVL in males, and having significantly higher ratios of IND, THL, TL, and FL to SVL in females (all p-values < 0.05; Table 1 and Table S6); from O. huanggangensis by having significantly lower ratios of NSL and IAE to SVL in males, having significantly higher ratios of IND, LAL, THL, and TFL to SVL in males, and having significantly higher ratios of LAL, THL, TL, and FL to SVL in females (all p-values < 0.05; Table 1 and Table S6).

Ecology

To present, O. kweichowensis sp. nov. has been found in three localities: Lengshuihe Nature Reserve (27°34′–27°26′N, 105°57′–106°04′E) in Jinsha Co., Meitan Co. (27°39′–27°42′N, 107°33′–107°35′E) and Zheng’an Co. (28°09′–28°20′N, 107°30′–107°38′E) in Guizhou Prov. of China. Geographical distances between these localities were from 89 to 173 km. Population from the Lengshuihe Nature Reserve inhabited broad streams, and near the riparian areas, surrounded by evergreen broadleaved forests (Fig. 10A). Populations from Meitan Co. and Zheng’an Co. inhabited broad slow-flowing rivers surrounded by paddy field (Figs. 10B and 10C). All of the localities were at elevations 717–766 m. All adult individuals that we found appear on the stones in the streams at night (07:30–12:00 pm) with water pH 6.8–7.1 and water temperature 15–23 °C. Tadpoles could be found at daytime and night. Amplexed individuals could be found in the streams in the type locality (Fig. 10A). Three sympatric amphibian species Fejervarya multistriata, Rana zhenhaiensis, and Polypedates megacephalus were found in Meitan Co. and Zheng’an Co., but only one sympatric amphibian species Amolops chunganensis was found in the Lengshuihe Nature Reserve in the type locality.

Figure 10 Habitats of Odorrana kweichowensis sp. Nov.

(A) Habitats in the type locality, Lengshuihe Nature Reserve, Jinsha County, Guizhou Province, China; insert is the photo for one pair of amplexed male (smaller) and female (larger) found on the stone in the stream. (B) Habitats in Xieba Town, Zheng’an County, Guizhou Province, China. (C) Habitats in Shilian Town, Meitan County, Guizhou Province, China. Photographs by S. Z. Li.

Etymology

The specific epithet “kweichowensis” refers to the distribution of this species, Guizhou Prov., China. The “kweichow” is an old spelling and a transliteration for “Guizhou.” We propose the common English name “Guizhou Odorous Frog” for this species.

Discussion

Morphological similarity of related species in the genus Odorrana challenges classifications in the group (Fei et al., 2009; Chen et al., 2013). Integrative taxonomy with multiple evidences, such as genetic divergence, ecological discrepancy, morphometric differences, and so on, turns out to be quite effective, and have already become the main trend. In this study, based on molecular phylogenetic analyses and morphological comparisons, a new species, O. kweichowensis sp. nov., was described from mountain streams in Guizhou Prov., China. Phylogenetic analyses based on mitochondrial DNA suggested that the new species belonged to Odorrana but was significantly separated from its congeners. Genetic distance based on 16S rRNA gene between it and its sister species O. schmackeri was mean 0.027, matching the level about interspecific divergences in amphibians (0.01–0.17; Vences et al., 2005) and being much higher than that between many sister species (of which, most species have been completely recognized as valid species) in Odorrana. Moreover, on nuclear protein-coding genes which presented much lower evolutionary rates, the new species was still indicated to occupy an independent lineage separated from other species. These results confirmed restricted and even absent gene flow between the new species and its related species. Finally, the new species was different from its congeners on a lot of morphological characters. Over all, multiple evidences supported the validity of the new species.

In this study, O. kweichowensis sp. nov. was found in three localities in the northwestern part of Guizhou Prov. of China (Fig. 1). In Li et al. (2015), the new species was only found in two localities in Guizhou Prov.: one was the type locality (Lengshuihe Reserve in Jinsha Co., Guizhou prov., China) of it, and another was the Suiyang Co., Guizhou prov., China closely adjoining Meitan Co., Guizhou prov., China and Zheng’an Co., Guizhou prov., China included in this study. Obviously, it could be speculated that the new species was at least distributed in the northwestern part of Guizhou Prov. of China (Fig. 1). Unfortunately, it was still not sure whether the new species was conspecific with Odorrana sp1 recognized by Zhu (2016) in absence of corresponding specimens and sequences. If they were conspecific, the new species should be distributed in a larger range from northern Guizhou Prov. to southern Guangxi Prov., China. In addition, it could not infer whether the populations in northern Thailand and northern Vietnam reported as O. schmackeri (Chan-ard, 2003; Orlov et al., 2002; Nguyen, Ho & Nguyen, 2005; Fig. 1) was conspecific with the new species, because there was also no molecular data and detailed morphological descriptions of them. As noted, according to previous studies (Li et al., 2015; Zhu, 2016) and our results, the new species was possibly sympatric with Clade D (Odorrana sp2) recognized by Zhu (2016) and O. hejiangensis in a narrow area along the boundary between northern Guizhou Prov. and Sichuan Prov. and Chongqing City, China (Fig. 1), though in the localities investigated in this study, we did not find the latter two species in the microhabitats of the new species.

Conclusion

We described a new species of the odorous frog genus Odorrana (Amphibia, Anura, Ranidae) from Guizhou Prov. of China, and provide evidence for its phylogenetic allocations. O. kweichowensis sp. nov. was only known from a narrow range in the northwestern part of Guizhou Prov. of China, and occurred from mountain streams at mid and low elevations similar to most odorous frogs. In our fieldwork, the new species was found to be seriously threatened by local villagers and construction of dams and roads. Thus, further more detailed investigations on the species are urgent to ascertain its distributional range and population status. With our description, we contributed to a better knowledge of the diversity of the genus Odorrana in the southwestern China, and thus suggested that more comprehensive phylogeographic studies would highlight radiation patterns of the group.

Supplemental Information

Supplemental Information 1 Table S1. Voucher information and measurements for the collected adult specimens in this study.

Unit: mm. Abbreviations for the morphometric characters refer to Methods section.

Click here for additional data file.

Supplemental Information 2 Table S2. Sampling number, voucher number, localities and GenBank Accession number for the sequence used in this study.

“*” denoted the sequences downloaded from GenBank.

Click here for additional data file.

Supplemental Information 3 Table S3. Primers used in this study.

Click here for additional data file.

Supplemental Information 4 Table S4. Measurements of the tadpoles of Odorrana kweichowensis sp. nov.

Unit: mm. Abbreviations for the morphometric characters refer to Methods section.

Click here for additional data file.

Supplemental Information 5 Table S5. Uncorrected p-distances between Odorrana species based on 16S rRNA gene sequences.

Click here for additional data file.

Supplemental Information 6 Table S6. Basic statistics for measurements of the adult specimens of Odorrana kweichowensis sp. nov., O. schmackeri and O. huanggangensis..

Unit: mm. Abbreviations for the morphometric characters refer to Methods section.

Click here for additional data file.

Supplemental Information 7 Raw data produced in this study.

Voucher information for each sequence refer to Table S2.

Click here for additional data file.

We thank C. Li and C. L. Liao on helping collection of samples. We are much indebted to the editor and referees for their valuable comments.

Additional Information and Declarations

Competing Interests

Author Contributions

Animal Ethics

Field Study Permissions

Data Availability

New Species Registration

The authors declare there are no competing interests.

Shize Li conceived and designed the experiments, performed the experiments, analyzed the data, contributed reagents/materials/analysis tools, prepared figures and/or tables, authored or reviewed drafts of the paper, approved the final draft.

Ning Xu contributed reagents/materials/analysis tools, authored or reviewed drafts of the paper, approved the final draft.

Jingcai Lv contributed reagents/materials/analysis tools, authored or reviewed drafts of the paper, approved the final draft.

Jianping Jiang contributed reagents/materials/analysis tools, authored or reviewed drafts of the paper, approved the final draft.

Gang Wei conceived and designed the experiments, analyzed the data, contributed reagents/materials/analysis tools, authored or reviewed drafts of the paper, approved the final draft.

Bin Wang conceived and designed the experiments, performed the experiments, analyzed the data, contributed reagents/materials/analysis tools, prepared figures and/or tables, authored or reviewed drafts of the paper, approved the final draft.

The following information was supplied relating to ethical approvals (i.e., approving body and any reference numbers):

The Animal Care and Use Committee of Chengdu Institute of Biology, CAS provided full approval for this purely observational research (Number: CIB2013041102).

The following information was supplied relating to field study approvals (i.e., approving body and any reference numbers):

Field experiments were approved by the Management Office of the Lengshuihe Nature Reserve (project number: LSH-201304003).

The following information was supplied regarding data availability:

The sequences described here are accessible via GenBank accession numbers MH193530–MH193617.

The following information was supplied regarding the registration of a newly described species:

Publication LSID: urn:lsid:zoobank.org:pub:E98B65CB-E8E3-4412-9613-D9DD32A77B99.

Species LSID (Odorrana odorrana kweichowensis): urn:lsid:zoobank.org:act:95123118-30D2-42B5-B5DB-AB7C32B33F97.

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
