# Peer review of "A new species of the odorous frog genus Odorrana (Amphibia, Anura, Ranidae) from southwestern China"

_PeerJ, doi:10.7717/peerj.5695_

## Round 0.1 · original submission · Major Revisions

Dear authors

Your ms has been reviewed. As you can see, our reviewers raised many questions. Please try to improve the ms accordingly. Be aware that we will send the revision to the original reviewers.

Kind regards,

Michael Wink
Academic editor

Reviewer 1 ·

Basic reporting

The language of the present study is understandable but substandard. The authors are requested to use professional language editing before resubmission.
No further issues.

Experimental design

I do not understand the rationale for some data analyses. Why were not all mitochondrial genes concatenated for phylogenetic analyses? Mitochondrial DNA behaves as one and the same locus and it makes little sense to run phylogenetic trees using 12S and 16S sequences, but leave out ND2 sequences and use these for another analysis (Splittree). I suggest to rerun the phylogenetic trees and Splittree using all genes concatenated.
In addition, I would have expected some considerations about the "barcoding gap" in frogs as the authors used genes, which are *the* barcoding genes in frogs and there is ample literature about this (e.g. Vences 2005, Phil. Transact. B 360: 1462; Vences et al. 2012, DNA Barcoding Amphibians and Reptiles, Springer; Murphy et al. 2013, Mol. Ecol. Res. 13: 161-167).
I have another major issue with the "Morphological comparisons" (starting with line 171). These morphological comparisons have in fact the character of differential diagnoses and are as such good. However, the "Diagnosis" (starting with line 229) is no diagnosis, but a description. A species diagnosis has to compare directly the new species with previously recognized ones, explaining that, for instance, species A differs from species B in this or that character. Of course species sharing character states may be lumped together here, but the diagnosis should lead the reader through the characters differentiating the new taxon from previously recognized ones.

Validity of the findings

The authors demonstrate all in all convincingly that they discovered a new frog species. However, the presentation (and language) are substandard and the paper needs to be rewritten in large parts.

Additional comments

This is a description of a new species of frog of the genus Odorrana -- a speciose and difficult genus. The material seems to be scientifically solid, however, I have some reservations regarding the publication of an individual species description in a journal like PeerJ. There are renowned journals (e.g., Zootaxa, ZooKeys) having their domain in the field of taxonomic descriptions, and I would see this paper rather there than here. In 2017 no less than 167 new frog species have been described (Kwet 2018, Elaphe 70) and for 2018 a similar figure is expected. The discovery of just one species description more does not qualify per se publication in a more generalist journal like PeerJ. I would have expected for such an outlet a more comprehensive revision of Odorrana, but not the description of only one new species.

Regardless of where the paper will end up, at the moment it suffers from several flaws making its acceptance impossible. Besides the issues highlighted above, there are many little issues. I pick just a few examples:

Line 300: it would be much more informative when the authors explain how far distant the three collection sites of the new species are.
Line 305: Etymology -- why is "kweichowensis" referring to the distribution of the new species in Guizhou? I assume that "Kweichow" is an old spelling/transliteration for "Guizhou". Right? If so, you have to explain this. As is, it is completely unclear.
Sentence starting with line 315 ("In recent years...") -- here is a reference lacking. Who has separated these species from O. schmackeri?
Table 1: "kweichowensis" is consistently misspelled as "keichouensis"

Reviewer 2 ·

Basic reporting

The English Language of the present manuscript requires a lot of improvement and checking, the paper must be proofread by an English native-speaking colleague. Introduction and Discussion are very short, I would prefer if the differentiation of Odorrana schmackeri complex would be introduced in more details. Structure follows PeerJ standards. Figures, however, are not acceptable at the current state and require a lot of polishing, especially the map (Fig. 1), phylogenetic trees and networks (Figs. 2 and 3) and PCA analysis results (Fig. 4).

Experimental design

Experimental design corresponds well to the objectives of the present paper. However, methods must be described in more details, for example tadpole identification and staging, osteological examinations etc. - this information is omitted.

Validity of the findings

The paper describes one of the lineages of Odorrana schmackeri species complex as a new species based on mtDNA, nuDNA and morphological data. However some statements (i.e. Guizhou province is a center of diversification of the genus Odorrana because it has 12 species recorded in it) appear to be speculative and unjustified and should be removed.

Additional comments

The manuscript is focused on taxonomic analysis of Odorrana schmackeri species complex based on data from mtDNA, nuDNA and morphology. From the data presented by the Authors, I am convinced the new species described in this manuscript indeed represents a new taxon which warrants taxonomic recognition. I congratulate the Authors on this discovery. It is especially important that the Authors applied an integrative approach and used both mtDNA and nuDNA genetic markers as well as morphological characters to assess diversity in this group of frogs. However, the way of data representation, especially figures, requires a lot of polishing and improvement. Also some issues on taxonomy of Odorrana need to be discussed in more detail prior to acceptance of this manuscript. Main issues to be addressed are listed below:
• Title. Your paper is titled “Odorrana kweichowensis sp. nov., a new species of the odorous frogs…”. This may be is not the most desirable title, since, according to the tradition in zoological taxonomy, it is not advised to use the new species name “prior” to publication of this name: technically all parts of your paper above the line where taxonomic description starts “Odorrana kweichowensis sp. nov.” are “before” the actual description of the new species. That’s why many journals, like many editors in Zootaxa or Zookeys for example, ask authors not to use the new species name before the line where taxonomic description starts, so all these new species should be cited as “Odorrana sp. 1” or similar way. Some editors even insist that the name should not be mentioned in Abstract or Keywords (which may be is not so useful). There is no strict rule on that, but I believe there is no need to give the new name in the title, I think the title “A new species of the odorous frogs (Odorrana, Amphibia, Anura, Ranidae) from the mountain streams of Guizhou Province of China” would be better for your paper.
• English. I strongly suggest that the Authors revise the English language of the manuscript with help of a English native speaker colleague or paid services for improvement of scientific writing. Right now it is clear that the manuscript was not even proofread before submission, I found numerous mistakes, typos and inaccuracies. The English language should be improved to ensure that an international audience can clearly understand your text; unfortunately this is not always the case in the present version of the manuscript. I am not a native speaker myself, but I suggest that the Authors pay special attention to consistent use of tenses throughout the manuscript. Some examples where the language could be improved are marked in the attached PDF. There are many so I don’t give details here.
• Introduction. Please always clearly denote what you mean by “Odorrana schmackeri species complex”. A reader can guess what species it includes only after carefully checking the tree. Please define it in the Introduction and provide a brief overview of its distribution and taxonomy.
• Introduction. Introduction is largely incomplete. Please discuss in more detail about Odorrana schmackeri complex - it's distribution range, previous dtata on taxonomic differentiation within this group of Odorrana, discuss results of molecular analyses of Li et al. 2015 and other authors. Please also mention why the followig species are considered to be the members of O. schmackeri complex: O. huanggangensis, O. yizhangensis and O. lungshengensis. Please provide also a review describing history of splitting of O. schmackeri complex; please discuss phylogenetic results of Li et al. 2015 in more detail. How the clades they revealed correspond to your clades and described members of the complex? They have much wider sampling than you and show more complicated phylogenetic paper… It is not clear what your new data bring to understanding of O. schmackeri complex differentiation compared to previous works.
• Materials and methods. Please specify how your sampling of O. schmackeri, O. huanggengensis, O. yizhangensis and O. lungshengensis correspond to the respective type localities of these species? Can we be 100% that your samples really include topotypic populations and represent these nominal taxa?
• Materials and methods. Why you call this research fully observational when you preserved specimens in formalin and ethanol? Please specify how you euthanized specimens. It is unclear in the present version of the paper.
• Materials and methods. How you prepared skull preparations? Were they stained with alizarin? How many specimens? Nothing is told on that in Materials and Methods, but you provide a drawing of a skull. Please add the relevant section.
• Molecular analyses. The statement on lines 87-88 puzzles me. You say “Mitochondrial gene segments could be concatenated into one partition because they were effectively inherited as one locus”. I don't agree with this. You combine non-coding - 12S and 16S and protein-coding ND2 genes, so absolutely you must test at least one partition for rRNA genes and three partitions for each codon for ND2 genes. In case you get same recommended models for both 12SrRNA +16SrRNA and all three partitions of ND2 you can join them in one partition. Also, 16S and 12S rRNA genes have usually many gaps in the alignment due to tertiary structure of rRNA molecules. How these gaps were treated in your analyses? Excluded or included? Please specify. I think you should re-run the Modeltest for more detailed partitioning scheme and redo the whole phylogenetic analyses.
• Morphological analyses. Please unify some morphological descriptions in the Diagnosis, Comparisons and holotype description. For example finger disks are regarded as “pointed”, “tapering”, “cordiform”, etc. It’s all quite confusing.
• Morphological analyses. Please provide references which you relied on when choosing morophological characters to examine!! Which method of measurement you followed?
• Morphological analyses. I am very confused - did you used the direct measurements for the PCA, as stated i this is unacceptable and cannot be taken seriously, since it doesn't exclude the body-size component from the analyzed variation. For proper taxonomic analysis you should: 1) Calculate ratio of each character to SVL. 2) Preform PCA analysis for character ratios or loge-transformed character ratios. This will give you real differences in body shape and proportions excluding the impact of differences in body size. Please redo this analysis.
• Morphological analyses. Same, when comparing morphometric characters between taxa (e.g. longer head VS shorter head) please compare character ratios, not actual measurements (except SVL). Please indicate what statistical tests you applied for testing the significance of differences you recorded.
• Molecular results. See Lines 167-170. Actually, 2.6-2.8% is not a high degree of divergence in frogs for 16S rRNA gene. Examples you show from other species of Odorrana are not always reliable since there may be mistake in identification of samles or incomplete taxonomy (such as the case of O. tiannanensis and O. bacboensis, most likely). Please refer to papers reviewing degree of variation of 16S rRNA gene in amphibians (see papers by Vences et al. 2006, Vieites et al., 2009 etc) and discuss how your values correspond to species-level divergence in frogs.
• Molecular results. Your strongest molecular argument is the concordance between nuDNA and mtDNA suggesting divergence of the linage you want to describe as a new species. This concordant phylogenetic signal is very important. However you do not stress or discuss this in your work. Please reconsider.
• Molecular results. You don't give actual node support values in the text of the MS, nor in Figure 2. Please add PP/BS values on each node for Figure 2.
• Morpological results. Lines 183-210. Most part of this section must be moved to Comparisons section in the new species description. Please expand and rewrite Comparisons section. Please first focus on differences from other Odorrana taxa and then describe in detail differences from O. schmackeri and all other "related" taxa of the same species group.
• Morpological results. The measurement accuracy you show in the present manuscript (with 2 decimals after dot) cannot be achieved in amphibian morphometrics. Please round ALL your measurements in the text and tables to one decimal after dot.
• Species description. Please spell new species name fully without abbreviations everywhere throughout the manuscript, including figures and tables.
• Species description. How was the species identity of tadpoles evaluated? DNA? Observation of development? Please specify, this should e explained in materials and methods. You staged tadpoles based on Gosner’s tables, but you decided not to cite his work. Please add Gosner’s publication to reference list.
• Species description. Diagnosis has to be reorganized. In Diagnosis first please give characters based on which the new species is attributed to the genus Odorrana: 1) ..., 2) ..., 3) .... etc. Then please list diagnstic characters for species compared with its congeners.
• Species description. When writing comparisons of morphometric characters between species or between sexes, please always provide RANGES of characters, not just some values. What are these values? Mean? What is the sample size? Better indicate it anywhere you compare two samples.
• Species description. Data on natural history of the new species are really scarce. Any idea about diet? Sympatric amphibian species? When and where were animals observed? Etc. Please expand.
• Discussion. Odorrana schmackeri is also reported for northern Thailand and even Vietnam (see Orlov et al. 2002 and Chan-ard et al. 2011). It has a big range (also please see Li et al. 2015). Please show the range of Odorrana schmackeri on the map in Figure 1. Please demonstrate on the map, how your taxonomic revision devides the range of the species - which populations will remain in O. schmackeri s. stricto, and which become your new species. Otherwise it is very difficult to understand for a reader. This is may be the crucial and most important problem of your paper.
• Discussion. The statement on the lines 320-321 that “the number of Odorrana species in Guizhou Province has increased from 8 to 12 (Fei et al., 2012), accounting for about 20.69% of total number of species in the genus. And because of that, Guizhou Province is considered to be the differentiation center of the genus Odorrana” appears to be completely unjustified. If you have 12 species instead of 8 in Guizhou province, you can not judge on evolutionary history of the whole genus and claim this is a diversification center! You need to do phylogeny and biogeography analysis for that. Please remove this.
• Discussion. I think the information you provided is not enough to judge that the new species is Vulnerable. It's rather Data Deficient since more surveys needed. If so, please give reference to IUCN criteria you used for conservation status estimation.
• Table 1. Please reorganize lines in table, may be better to sort according to species names? Also in this table you use a completely different spelling of the new species as “keichouensis”. Please correct that!
• Table 2. All measurements should be given to first decimal.
• Table 3. All measurements should be given to first decimal.
• Figure 1. This drawing must be reorganized completely: 1) Please provide a map showing elevation (since Odorrana are found in montane areas this is important). 2) Please show the known range of all members of Odorrana schamckeri species complex, inclduing O. schmackeri s. stricto and the know records from Thailand etc. 4) Please provide color figures for localities which would correspond to colors used in Figures 2 and 3. Otherwise it is very difficult to recognize which locality is which species. 5) Please provide legend to locality names and species names to this figure.
• Figure 2. This figure is difficult to read and needs to be polished. I suggest it's better to separate it into 2 figures. 1) For the 12S-16S tree please make font size bigger. 2) Please use colors which would help to distinguish species of Odorrana schmackeri complex. 3) Please add 12S 16S genetic data for O. yizhangensis and O. lungshengensis which are missing from the 12S 16S tree. 4) Please provide actual node support values for BI/ML analyses, it is much more useful than these circles indicating maximal support everywhere. 5) Please mark the monophyletic O. schmackeri complex and it's members. For ND2 network 1) Better make it a separate figure; 2) Use same colors as in the 12S16S tree and Figure 1.
• Figure 3. This figure also needs editing. Please use same colors as used for marking clades in Figures 1 and 2. If possible please could you choose more bright colors (green, red, yellow, blue), for color-blind people might be difficult to distinguish the colors you used. For me it's difficult to distinguish between colors of yizhangensis and lungshengensis, for example.
• Figure 4. This figure must be redone completely. 1) recalculate plots for charcater ratios to SVL, not actual measurement values, to avoid the effect of differences in body size. 2) Please mark that A if for males, B is for females (or which one is which?). 3) You cannot draw borders of polygons like that! You should show minimal CONVEX polygon for each species, you cannot draw borders of these figures as you like. When you plot convex polygons and see how they overlap this shows the degree of differences between samples you compare. Please redo!
• Figure 8. You need to add signatures for the skeletal features you mention in the respective part of the manuscript and a legend to this figure.
• Figure 9. There is no “formula” shown in this figure.

Annotated reviews are not available for download in order to protect the identity of reviewers who chose to remain anonymous.

---

## Round 0.2 · Minor Revisions

Dear authors,

Your revision has improved your ms but I have a few items to be changed:

Collection dates; the hours are less important than the actual dates, at least years; provide details

Fig 1. coloured names are difficult to read; put them in a coloured box or improve otherwise

S5: In English, you use a dot instead of comma for decimal values

Your ms would profit from being edited by a native English speaker

Hope that you can revise your ms accordingly

Greetings
Michael Wink
AE

Reviewer 2 ·

Basic reporting

The current version of the manuscript is greatly improved, the English is also much more comprehensible. I believe the present version of the manuscript can be accepted for publication.

Experimental design

All experiments are described correctly with sufficient details.

Validity of the findings

Presented data are important for taxonomy of Odorrana, the authors report on a new species of the genus.

Additional comments

I thank the authors for their hard work on the manuscript and suggest that the present version can be accepted for publication.

---

## Round 0.3 · accepted · Accept

Dear authors,

Congratulations! Your revision is adequate and now you ms is ready for the final editing by our staff.

Thank you for submitting your work to our journal. xie xie

Regards
Michael Wink
Academic Editor

#